# Differentiation of Intracerebral Tumor Entities with Quantitative Contrast Attenuation and Iodine Mapping in Dual-Layer Computed Tomography

**DOI:** 10.3390/diagnostics12102494

**Published:** 2022-10-15

**Authors:** Jan Borggrefe, Max Philipp Gebest, Myriam Hauger, Daniel Ruess, Anastasios Mpotsaris, Christoph Kabbasch, Lenhard Pennig, Kai Roman Laukamp, Lukas Goertz, Jan Robert Kroeger, Jonas Doerner

**Affiliations:** 1Institute for Radiology, Neuroradiology and Nuclear Medicine, Johannes Wesling University Hospital by Muehlenkreiskliniken, Ruhr University Bochum, 32429 Minden, Germany; 2Department of Radiology and Nuclear Medicine, Universitätsmedizin Mannheim, 68167 Mannheim, Germany; 3Institute for Diagnostic and Interventional Radiology, University Hospital Cologne, 50397 Cologne, Germany; 4Department of Stereotaxy und Functional Neurosurgery, University Hospital of Cologne, 50937 Cologne, Germany; 5Department of Neuroradiology, München Kliniken Harlaching, 81545 München, Germany

**Keywords:** neurooncology, tumor differentiation, iodine quantification, dual-energy computed tomography

## Abstract

Purpose: To investigate if quantitative contrast enhancement and iodine mapping of common brain tumor (BT) entities may correctly differentiate between tumor etiologies in standardized stereotactic CT protocols. Material and Methods: A retrospective monocentric study of 139 consecutive standardized dual-layer dual-energy CT (dlDECT) scans conducted prior to the stereotactic needle biopsy of untreated primary brain tumor lesions. Attenuation of contrast-enhancing BT was derived from polyenergetic images as well as spectral iodine density maps (IDM) and their contrast-to-noise-ratios (CNR) were determined using ROI measures in contrast-enhancing BT and healthy contralateral white matter. The measures were correlated to histopathology regarding tumor entity, isocitrate dehydrogenase (IDH) and MGMT mutation status. Results: The cohort included 52 female and 76 male patients, mean age of 59.4 (±17.1) years. Brain lymphomas showed the highest attenuation (IDM CNR 3.28 ± 1,23), significantly higher than glioblastoma (2.37 ± 1.55, *p* < 0.005) and metastases (1.95 ± 1.14, *p* < 0.02), while the differences between glioblastomas and metastases were not significant. These strongly enhancing lesions differed from oligodendroglioma and astrocytoma (Grade II and III) that showed IDM CNR in the range of 1.22–1.27 (±0.45–0.82). Conventional attenuation measurements in DLCT data performed equally or slightly superior to iodine density measurements. Conclusion: Quantitative attenuation and iodine density measurements of contrast-enhancing brain tumors are feasible imaging biomarkers for the discrimination of cerebral tumor lesions but not specifically for single tumor entities. CNR based on simple HU measurements performed equally or slightly superior to iodine quantification.

## 1. Introduction

Brain tumors are classified into a variety of subtypes that have various molecular characteristics. different clinical appearances, treatment possibilities, and patient outcomes [1]. The pathophysiology of brain tumors is associated with many factors and varies between brain tumor lesions of primary and secondary origin as well as local host factors [2]. Tumor metabolism varies with the function and architecture of tumor blood vessels and therefore also between the different tumor entities [3,4]. For patients with brain tumors, it is particularly important to determine the treatment scheme and evaluate the survival prognosis [5,6].

With the goal to provide this guide to treatment and survival estimation, current research in computed tomography (CT) and magnetic resonance imaging (MRI) aims to characterize brain tumors noninvasively [6,7,8,9,10]. For this purpose, texture and density features, radiomics and/or deep learning are used on imaging data in order to provide standardized in-depth information on tumor characteristics and tumor entity [6,11,12,13,14]. In clinical imaging, magnetic resonance imaging (MRI) with and without gadolinium-based contrast agents is the imaging method of choice. However, computed tomography has dedicated strengths in comparison to MRI, providing quantitative measures of tumor-associated brain blood barrier breakdown and neovascularization [1,6,15].

Conventional computed tomography (CT) relies on the attenuation of different tissues using a single X-ray spectrum. The introduction of dual-energy CT (DECT) has allowed for more sophisticated material separation algorithms using acquisitions with different tube voltage settings. In recent years, detector-based approaches such as photon counting CT and dual-layer dual-energy CT (dlDECT) (IQon, Philips, Best, The Netherlands) were introduced that allow for more accurate and distinct quantification of iodine in between scans and scanners, as they make use of the dedicated distribution of attenuation values measured in Hounsfield units (HU) as a function of X-ray energy [16,17]. This allows the scan to be decomposed into clinically relevant materials such as iodine, calcification, uric acid, and soft tissues. There are different technical source-based approaches such as dual-source or rapid kilovoltage switching. dlDECT, as the method of choice for DECT in this work, uses a single polychromatic x-ray source. The separation of different X-ray energies with dlDECT occurs at the detector level. It detects photons of lower energy in the upper level and higher energy photons in the layer below [18,19]. Current systems provide spectral image data with every scan, without the need for additional radiation exposures.

Although there may be distinct advantages of DECT in regard to the evaluation of blood–brain barrier breakdown in neurooncological cases, literature is extremely sparse in regard to the value of CT in the differentiation of brain tumors which is likely associated with the predominance of MRI and a lack of standardization of examination protocols [6]. Examining a unique dataset of patients receiving highly standardized dlDECT for the purpose of stereotactic biopsies, we aimed to evaluate the value of tumor attenuation and iodine uptake of brain tumor lesions as a potential quantitative biomarker for the discrimination of tumor entities. Further, we aimed to test if iodine maps improve tumor separation in comparison to attenuation measurements on conventional polyenergetic reconstructions of the same scans.

We hypothesized that quantitative CT attenuation measures in dlDECT allow the differentiation of brain tumor entities and tested conventional attenuation measurements as well as iodine mapping for this task. Glioblastomas, brain metastases, and lymphomas, which are common brain malignancies in adults and which tend to show similar enhancement patterns in magnetic resonance imaging (MRI) [20], were of dedicated interest. Further, we evaluated the association of quantitative contrast enhancement with prognostic molecular markers such as isocitrate dehydrogenase (IDH) and O6-methylguanine-DNA-methyltransferase (MGMT) status in gliomas [8,21].

## 2. Materials and Methods

### 2.1. Study Population

Institutional review board approval was obtained for this study. The retrospective study includes CT data of all consecutive 139 patients who were referred to our center for CT examination before stereotactic needle biopsy between January 2017 and April 2018 and who received dlDECT (IQon Spectral CT, Philips Healthcare, Best, The Netherlands) receiving a dedicated scanning protocol (Figure 1). Besides imaging parameters, patient age, weight, gender, and histopathological diagnosis were recorded. If there were less than five tumors for a dedicated tumor entity or if the results in histopathology were inconclusive patients were excluded from further analysis.

### 2.2. CT Scans

All patients were scanned with the same dlDECT (IQOn, (Philips Healthcare, Best, The Netherlands). CT scans were conducted with a defined dosage of 60 mL contrast media (Accupaque 350 mg/mL, GE Healthcare; Little Chalfort, UK) and scan 30 seconds after bolus application via an antecubital vein, 4 mL/s flow rate, followed by the saline chaser. The following scanning parameters were kept constant in all scans: collimation = 64 × 0.625 mm; rotation time = 0.4 s; pitch = 0.422; tube potential = 120 kVp, matrix = 512 × 512; 300 mAs without dose modulation. Patients were positioned supine and scanned in a craniocaudal direction. All axial images were reconstructed with a slice thickness of 2 mm and a section increment of 1 mm using a dedicated spectral reconstruction algorithm with a strength level of 3 and a constant kernel (Spectral B, Philips Healthcare, Best, The Netherlands). Image analysis was performed offline on a dedicated workstation (IntelliSpace Portal 6.5, Philips Healthcare, Best, The Netherlands).

### 2.3. Objective Image Analysis

Datasets were analyzed by a blinded reader placing circular regions of interest (ROIs) in the contrast-enhancing part of the lesion and healthy appearing white matter at the contralateral side. Absolute attenuation values and standard deviations in Hounsfield units (HU) were recorded. Further, iodine density mapping was provided with the use of the manufacturers’ post-processing (IntelliSpace Portal V) and extracted via ROI measurements by a further blinded reader. Measurements were performed twice and averaged. Contrast to noise ratios (CNR) were calculated using the following formula: CNR = (HU_lesion_ − HU_contralateral_)/√(SD_lesion_^2^ + SD_contralateral_^2^)(1)

### 2.4. Stereotactic Needle Biopsy

A stereotactic needle biopsy was performed on the day of the CT examination and conducted by the department of stereotactic neurosurgery at our center. Pathologic work-up was provided as a reference standard for each lesion.

### 2.5. Statistical Analysis

Statistical analysis was performed using JMP (V15; SAS Institute, Cary, NC, USA). Student’s *t*-test, Wilcoxon test and Pearson correlation coefficient were used to compare the continuous variables. The association of iodine density and brain tumors as well as iodine thresholds were evaluated by logistic regression analysis, followed by receiver operating characteristics (ROC). Statistical significance was set to *p* ≤ 0.05. Further results are summarized as mean ± standard deviation.

## 3. Results

The mean age was 59.4 ± 17.1 years. The cohort included 52 female and 76 male patients. Histopathological diagnoses were: WHO grade II astrocytoma (n = 6), WHO III anaplastic astrocytoma (n = 23), glioblastoma multiforme (GBM, n= 62), CNS-Lymphoma (n = 12), metastasis (n = 15), oligodendroglioma (n = 5) and pilocytic astrocytoma (n = 5). Following tumors were excluded due to low incidence: germinoma (n = 1), ependymoma (n = 1), WHO I angiocentric glioma (n = 1), craniopharyngioma (n = 2), myeloid sarcoma (n = 1). Inconclusive histopathology was present in two cases.

Lymphomas showed the strongest CNR (3.28 ± 1.23), significantly higher than glioblastoma (2.37 ± 1.55, *p* < 0.005, Figure 1) and metastases (1.95 ± 1.14, *p* < 0.02), which did not differ from each other significantly (ns). Pilocytic astrocytomas had a comparable iodine density to metastases, lymphoma and GBM (Table 1). These four strong enhancing lesion types differed significantly from the oligodendroglioma, Grad II and Grad III astrocytoma that showed IDM CNR in the range 1.22–1.27 (±0.45–0.82). 

CT attenuation and iodine density differed statistically significantly between different tumor etiologies (Table 1, Figure 2). In comparison to simple attenuation or iodine density measures, the calculation of CNR improved tumor differentiation considerably. Neither in the simple attenuation and iodine density measurements nor in the CNR measurements did iodine density outperform conventional CT attenuation. 

In the subset of 78 gliomas, CNR was significantly higher in IDH 1 wild-type gliomas than in gliomas with IDH 1 mutation, both in conventional polyenergetic CT (2.19 ± 1.33 vs. 1.39 ± 1.4, *p* = 0.004) and iodine density mapping (2.22 ± 1.53 vs. 1.60 ± 1.78, *p* = 0.03) (Figure 3). Here, conventional polyenergetic CT attenuation measures performed better than density measures in iodine density maps regarding IDH mutation status as well. MGMT methylated gliomas showed a lower density in comparison to MGMT unmethylated gliomas which was however not significant (*p* = 0.11–0.22).

Tumor density was a significantly higher in women (62.2 ± 22.1 HU) in comparison to men (54.9 ± 19.9 HU, *p* = 0.04, Figure 4). In contrast, there were no associations of tumor enhancement with patient age and weight (both r^2^ = 0.02, ns, Figure 5a,b). 

## 4. Discussion

The preoperative determination of histopathology with imaging is a current goal of cancer research. Different AI algorithms and studies using radiomics investigated, how to separate tumor entities with imaging but there are no larger studies, protocols or products that allow for a secure differentiation of brain tumor lesions. This study investigated quantitative density measures in dual-layer computed tomography as potential biomarkers for the differentiation of brain tumors. Further, the study compared conventional polyenergetic algorithms and dlDECT iodine mapping for this purpose. In addition, CT measures were tested for differentiation of IDH mutation status and MGMT status in the subset of gliomas.

We found that CNR of brain tumor attenuation and healthy appearing contralateral white matter was superior for tumor discrimination than attenuation measures within the tumor only (Figure 2a,b). CNR of standardized iodine mapping allowed for a statistical separation of brain tumor entities. However, there was no improvement in comparison to conventional CT measures in iodine maps derived from dlDECT. Thus, we infer that there is no dedicated advantage to using specific dual energy iodine maps for this purpose and conventional computed tomography with specific protocols may yield similar results. In view of the recent results by Yingying et al., our study confirms the differences shown for lower and higher-grade gliomas [6]. However, looking at a broader range of brain tumors, we found that CNR did not allow any discrimination of glioblastoma from single brain metastases, which is a relevant differential diagnosis [22,23]. In contrast, brain lymphomas differed significantly showing the strongest quantitative enhancement of all brain tumor entities included.

Further, CNR of both, iodine density measurements and conventional CT measurements allowed for a separation of gliomas with IDH mutation and IDH wild type as well as MGMT promoter status. These findings are of particular interest, as these factors are strongly correlated with patient outcomes [8,21,24,25]. Here, CT density measures as one biomarker allowed for a relative differentiation of genetic status in this regard, however, did not yield a strong predictive value as an imaging biomarker. In view of further tumor characteristics defined in imaging with radiomics or AI, iodine density CNR measures could potentially play a role in non-invasive tumor differentiation and follow-up [22,26,27,28].

The cohort that received standardized planning dual-layer CT for stereotaxis of non-treated lesions with profound matching histological analysis is unique. Still, the study has limitations. It is a retrospective study with a limited amount of data for single tumor entities. The data were acquired with one single CT scanner and time point, so that reproducibility between scanners and time points remains to be tested. Further, there is a need for a dedicated evaluation of confounding factors such as contrast protocols, cardiac output function and gender. Patient weight and age did not affect the results of our study.

We conclude that attenuation and quantitative iodine density mapping of brain tumors allow for the statistical separation of brain tumor subtypes. In combination with advanced image analysis with deep learning and/or radiomics, the approach could potentially allow for non-invasive tumor classification. CNR based on simple attenuation measures performed equally or slightly superior to dual-energy iodine quantification so that the specific dual-energy function was not of added value. CNR of both, iodine density measurements and conventional CT measurements allowed for a separation of gliomas with IDH mutation and IDH wild type.

## Figures and Tables

**Figure 1 diagnostics-12-02494-f001:**
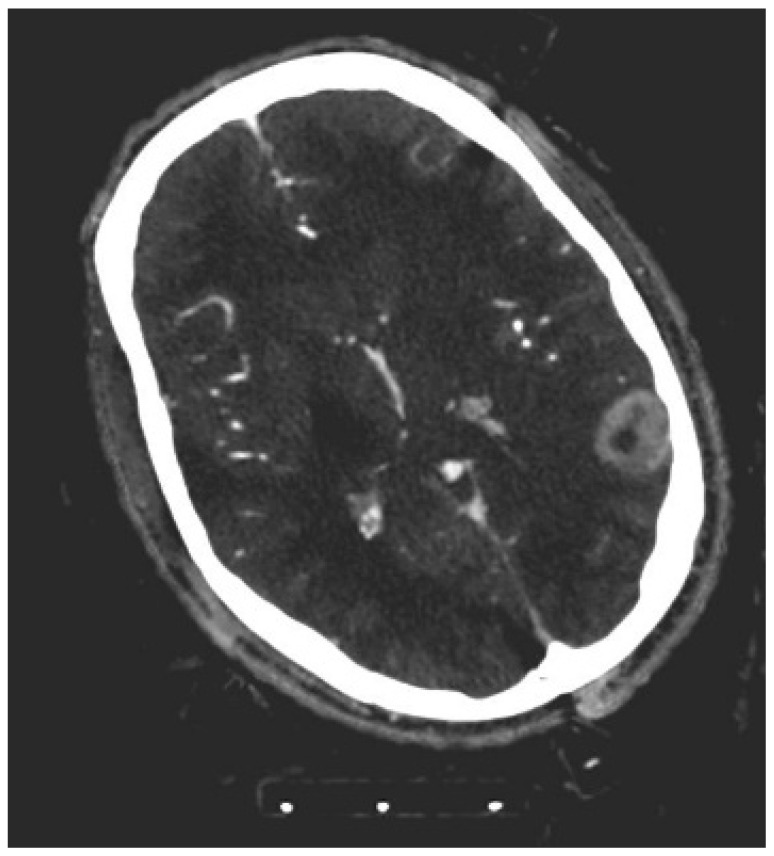
Iodine map of dual layer computed tomography showing a contrast-enhancing glioblastoma (WHO IV) located in the left-sided temporooccipital gyrus.

**Figure 2 diagnostics-12-02494-f002:**
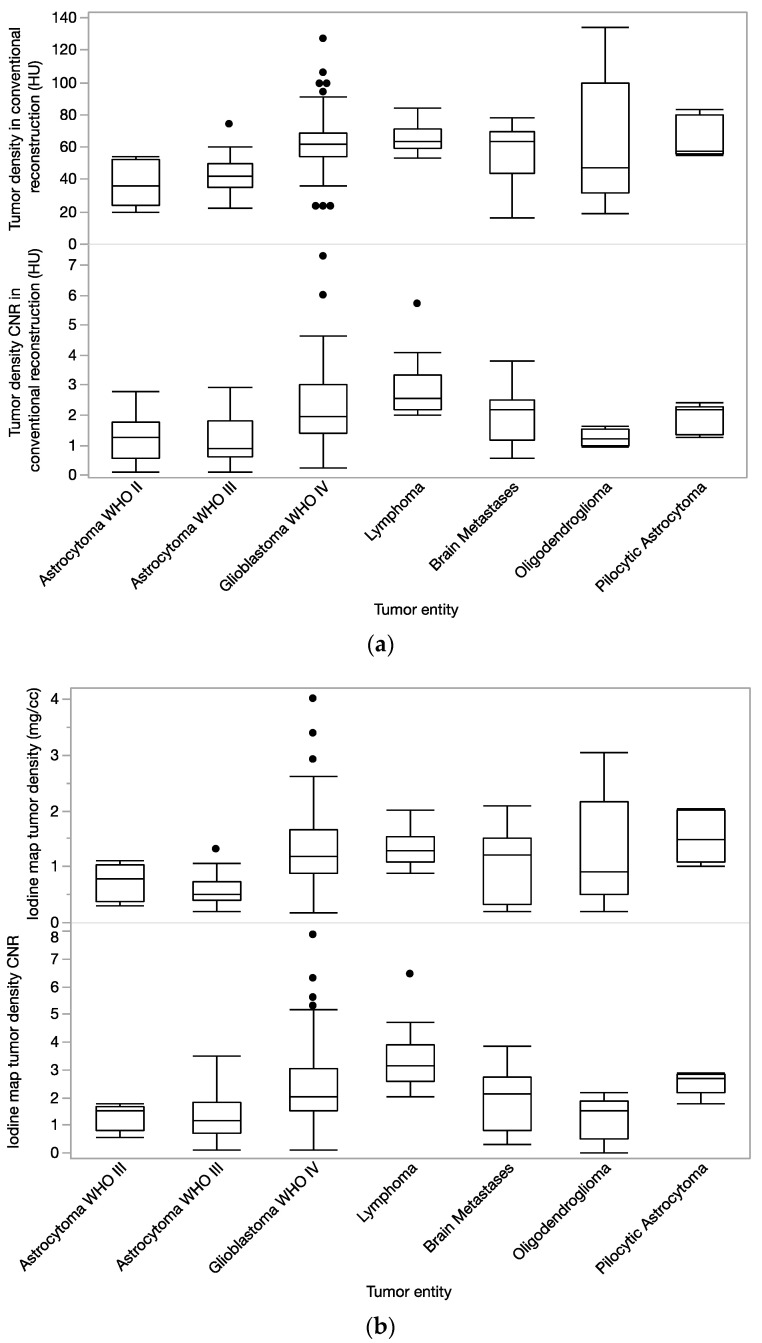
(**a**,**b**): Differentiation of tumor entity based on density measures and CNR of contrast enhancement in conventional polyenergetic reconstructions of DLCT (Figure 1a) and DLCT iodine maps. In contrast to simple density measures, CNR allows for improved differentiation between the prevalent tumor entities (compare data listed in Table 1). Lymphoma showed the highest CNR, significantly higher than the second strongest enhancing tumor entities glioblastoma and brain metastases which did not show a significant difference in tumor density. The astrocytomas WHO II and WHO III as well as oligodendrogliomas showed a significantly lower CNR than lymphomas, glioblastomas and metastases.

**Figure 3 diagnostics-12-02494-f003:**
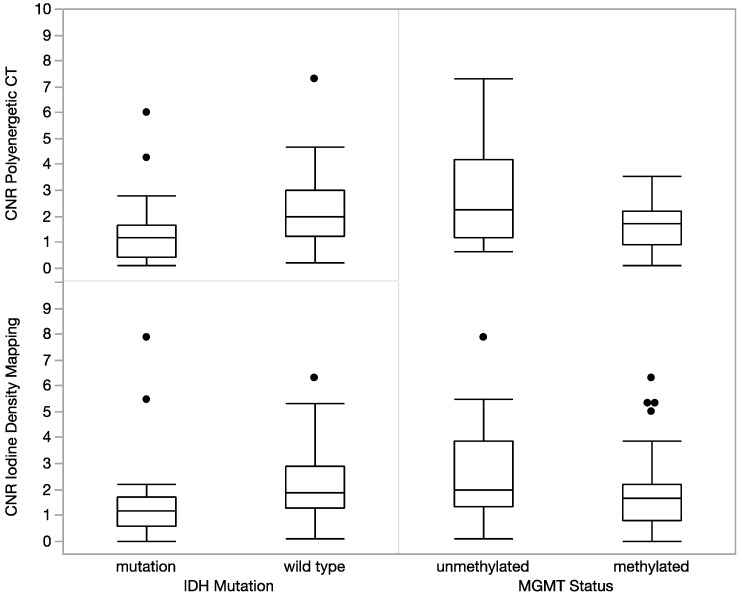
In the subset of 78 gliomas, CNR was significantly higher in IDH 1 wild-type gliomas than in gliomas with IDH 1 mutation, both in conventional polyenergetic CT (2.19 ± 1.33 vs. 1.39 ± 1.4, *p* = 0.004) and iodine density mapping (2.22 ± 1.53 vs. 1.60 ± 1.78, *p* = 0.03). MGMT methylated gliomas showed a lower density in comparison to MGMT unmethylated gliomas which was however not significant (*p* = 0.11–0.22).

**Figure 4 diagnostics-12-02494-f004:**
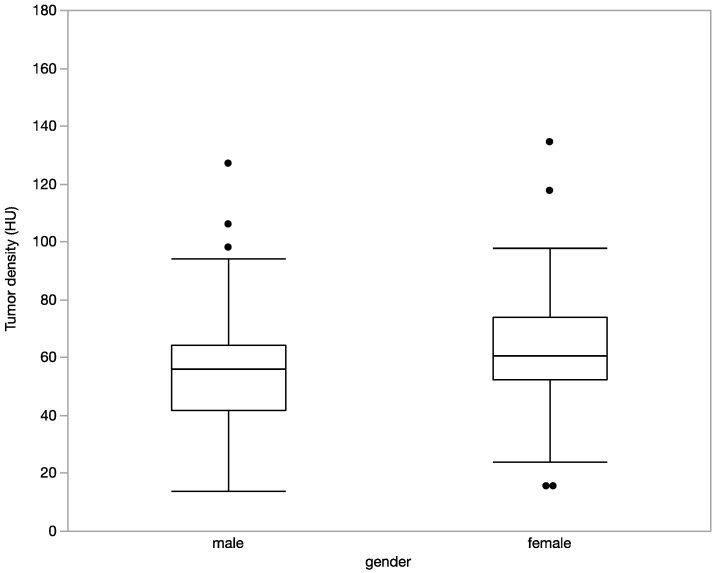
There was a significantly higher tumor density in women (62.2 ± 22.1 HU) in comparison to men (54.9 ± 19.9 HU, *p* = 0.04).

**Figure 5 diagnostics-12-02494-f005:**
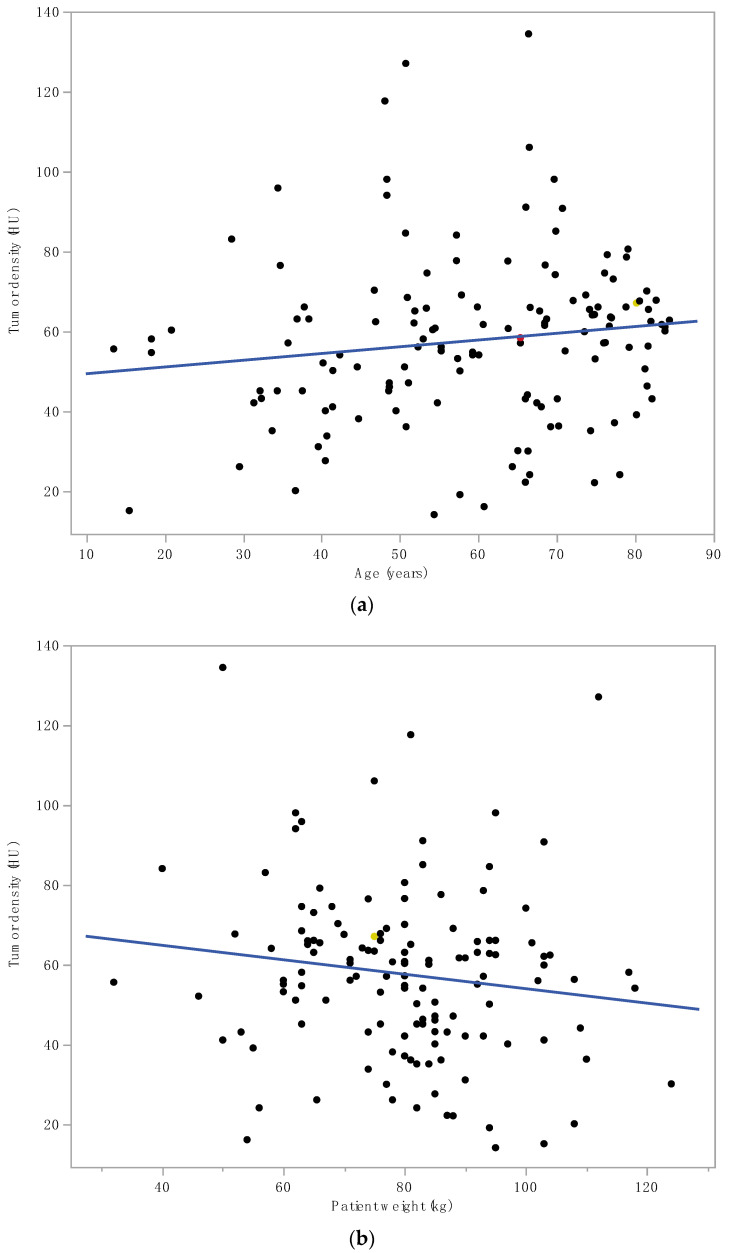
There was no significant correlation between tumor density and patient (**a**) age (r^2^ = 0.02, ns); (**b**) weight (r^2^ = 0.02, ns).

**Table 1 diagnostics-12-02494-t001:** Comparison of conventional CT and iodine mapping density measures and CNR for the discrimination of brain tumor entities (BTE) depicted with Wilcoxon z-scores and r-values. There is no significant difference between the conventional CT and the iodine mapping which provided both measures for significant differentiation of seven tumor entities. CNR was of dedicated advantage allowing for significant differentiation of 10 tumor entities, both in conventional CT as well as iodine mapping. (ns = not significant).

		Conventional CT Density		CNR Conventional CT Density		SDCT Iodine Density		CNR Iodine Density	
BTE A	BTE B	Wilcoxon Z-Score	Wilcoxon r-Value	Wilcoxon Z-Score	Wilcoxon r-Value	Wilcoxon Z-Score	Wilcoxon r-Value	Wilcoxon Z-Score	Wilcoxon r-Value
Astrocytoma II	Pilozytic Astroytoma	2.6	<0.01	1.2	ns	2.5	<0.05	2.6	<0.05
Astrocytoma II	Lymphoma	3.2	<0.01	2.7	<0.01	2.9	<0.01	3.3	<0.001
Astrocytoma II	Glioblastoma	3.1	<0.01	2.2	<0.05	2.1	<0.05	2.0	<0.05
Astrocytoma II	Oligodendroglioma	0.8	ns	0.1	ns	0.6	ns	0.1	ns
Astrocytoma II	Metastases	2.6	<0.01	1.4	ns	0.7	ns	1.3	ns
Astrocytoma II	Astrocytoma III	0.8	ns	0.1	ns	1.0	ns	1.3	ns
Astrocytoma III	Pilozytic Astroytoma	3.0	<0.01	1.7	ns (*p* = 0.08)	3.2	<0.01	2.6	<0.01
Astrocytoma III	Lymphoma	4.2	<0.0001	4.1	<0.0001	4.4	<0.0001	4.3	<0.0001
Astrocytoma III	Metastases	2.6	<0.01	2.7	<0.01	2.1	<0.05	1.9	ns (*p* = 0.06)
Astrocytoma III	Glioblastoma	4.7	<0.0001	3.7	<0.001	4.6	<0.0001	3.6	<0.001
Astrocytoma III	Oligodendroglioma	1.1	ns	0.4	ns	1.5	ns	0.4	ns
Lymphoma	Metastases	0.6	ns	2.0	<0.05	1.1	ns	2.8	<0.01
Lymphoma	Oligodendroglioma	1.1	ns	3.1	<0.01	1.3	ns	3.0	<0.01
Lymphoma	Glioblastoma	0.8	ns	2.3	<0.05	0.7	ns	2.9	<0.01
Lymphoma	Pilozytic Astroytoma	0.7	ns	2.0	<0.05	0.8	ns	1.3	ns
Glioblastoma	Oligodendroglioma	0.8	ns	2.3	<0.05	0.7	ns	1.8	ns (*p* = 0.06)
Glioblastoma	Metastases	0.1	ns	0.2	ns	0.9	ns	0.5	ns
Glioblastoma	Pilozytic Astroytoma	0.2	ns	0.3	ns	1.1	ns	1.1	ns
Pilozytic Astroytoma	Oligodendroglioma	0.8	ns	1.9	ns (*p* = 0.06)	1.0	ns	2.3	<0.05
Pilozytic Astroytoma	Metastases	0.3	ns	0.2	ns	1.3	ns	1.2	ns
Oligodendroglioma	Metastases	0.3	ns	1.7	ns (*p* = 0.08)	1.0	ns	1.2	ns

## Data Availability

The data are available from the corresponding author with reasonable requests.

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
