# Peer review of "Differentiation of Intracerebral Tumor Entities with Quantitative Contrast Attenuation and Iodine Mapping in Dual-Layer Computed Tomography"

_diagnostics, 2022, doi:10.3390/diagnostics12102494_

Round 1

Reviewer 1 Report

The paper is interesting.

Some suggestions/corrections:

1) Ln 11: definition of DLCT is missing. Please add it;

2) The sequence of figures is not correct. Please insert figures in the correct sequence;

3) There is only one figure that shows contrast enhancing glioblastoma in iodine map of dual layer computed tomography. I encourage authors to add some emblematic figures with captions of others brain tumors histopathological diagnosed in this study (WHO grade II astrocytoma, WHO III anaplastic astrocytoma, glioblastoma multiforme, CNS-Lymphoma, metastasis, oligodendroglioma and pilocytic astrocytoma).

Author Response

The paper is interesting.

Some suggestions/corrections:

  • Ln 11: definition of DLCT is missing. Please add it;

Answer: please excuse this error – dlDECT replaced DLCT in the process of writing and this one DLCT remained. We changed the text to dlDECT which is defined in the text and abstract.

  •  The sequence of figures is not correct. Please insert figures in the correct sequence;

Answer: the sequence of the images as it appears in the text was edited by the journal. The figure numbers are in line with there sequence of reference as it appears in the text. If required, I would like to ask the journal editors to change the figures in their sequence (1-5). However, I would also be OK with the figures remaining in the chosen sequence, as it has been suggested for best understanding of the reader.

  • There is only one figure that shows contrast enhancing glioblastoma in iodine map of dual layer computed tomography. I encourage authors to add some emblematic figures with captions of others brain tumors histopathological diagnosed in this study (WHO grade II astrocytoma, WHO III anaplastic astrocytoma, glioblastoma multiforme, CNS-Lymphoma, metastasis, oligodendroglioma and pilocytic astrocytoma).

Answer: As the images of the CT are not specific for the diagnosis, an as the measured quantitative data is in the main focus of the work, we would like to remain with the given image.

Reviewer 2 Report

The manuscript “differentiation of intracerebral tumor entities with quantitative contrast attenuation and iodine mapping in dual-layer computed tomography” reports an interesting study on quantitative contrast enhancement and iodine mapping of common brain tumor (BT) entities. The study is based on 139 consecutive standardized dual-layer dual-energy CT (dlDECT) scans conducted prior to stereotactic needle-biopsy of untreated primary brain tumor lesions.

1.     Line 106-109, “The following scanning parameters were kept constant in all scans: collimation – 64 × 0.625 mm; rotation time – 0.4 s; pitch – 0.422; tube potential – 120 kVp, matrix – 512 × 512; 300mAs without dose modulation.” I would rewrite this statement as “the following scanning parameters were kept constant in all scans: collimation = 64 × 0.625 mm; rotation time = 0.4 s; pitch = 0.422; tube potential = 120 kVp, matrix = 512 × 512; 300 mAs without dose modulation.”

2.     Figure 1, the glioma region should be indicated in the image.

3.     Discussion is shallow;

 Line 232, what are confounding factors which authors think to be evaluated?

 Line 233, repeated one.

Authors should discuss on figure 5 findings with suitable references.

Author Response

The manuscript “differentiation of intracerebral tumor entities with quantitative contrast attenuation and iodine mapping in dual-layer computed tomography” reports an interesting study on quantitative contrast enhancement and iodine mapping of common brain tumor (BT) entities. The study is based on 139 consecutive standardized dual-layer dual-energy CT (dlDECT) scans conducted prior to stereotactic needle-biopsy of untreated primary brain tumor lesions.

  1. Line 106-109, “The following scanning parameters were kept constant in all scans: collimation – 64 × 0.625 mm; rotation time – 0.4 s; pitch – 0.422; tube potential – 120 kVp, matrix – 512 × 512; 300mAs without dose modulation.” I would rewrite this statement as “the following scanning parameters were kept constant in all scans: collimation = 64 × 0.625 mm; rotation time = 0.4 s; pitch = 0.422; tube potential = 120 kVp, matrix = 512 × 512; 300 mAs without dose modulation.

Answer: We replaced the text as suggested. – It is marked in yellow. Thank you for this improvement!

       2. Figure 1, the glioma region should be indicated in the image.

Answer: We adapted the text: „Iodine map of dual layer computed tomography showing a contrast enhancing glioblastoma (WHO IV) located in the left sided temporooccipital gyrus.“

        3. Discussion is shallow;

 Line 232, what are confounding factors which authors think to be evaluated?

Answer: In regard to the confounding factors, this is correct. Our current study did not have the power to evaluate possible confounding factors in depth. However, we addressed this question and found that there were minor confounding effects associated to gender, however none associated to age and weight. We believe that this is of scientific interest, especially in the current research in MRI that is aiming to allow for non-invasive discrimination of different brain tumor entities. There are currently no studies that we know of, that evaluate the confounding factors of tumor contrast enhancement of the brain.

We adapted the paper: „Further, the study did not allow for a dedicated evaluation of confounding factors such as contrast protocols, cardiac output function and gender by tumor entity.“

   Line 233, repeated one.

Authors should discuss on figure 5 findings with suitable references.

Answer: Yes, there could be indeed more discussion on the subject. We performed another literature research to look for suitable references. However, the data on quantitative tumor enhancement is extremely sparse, especially in regard to the aspect of gender differences. Do you have suitable references that you would suggest? – Else, we would recommend the following wording that we included in the text:

„In our dedicated research of literature however, we did not find data on confounding factors for quantitative brain tumor enhancement in the field of MRI and CT. Thus, although it is beyond the scope and statistical power of our study, we performed statistical evaluation for possible confounding factors within our study data. Here, we found that there was a minor but statistically significant effect of gender (Fig. 5). In contrast, patient weight and age did not affect the results in our study. These observations should be evaluated in future studies that evaluate quantitative tumor enhancement in general.“

Round 2

Reviewer 2 Report

The revised version of the manuscript response to my comments partially which is not satisfactory. However, I understand the limitation of the study and as considering the importance of the finding, I would recommend for publication of this article.